# Heat Transfer of Magnetohydrodynamic Stratified Dusty Fluid Flow through an Inclined Irregular Porous Channel

**DOI:** 10.3390/nano12193309

**Published:** 2022-09-23

**Authors:** Gajendran Kalpana, Salman Saleem

**Affiliations:** 1Department of Mathematics, Mount Carmel College, Bengaluru 560 051, India; 2Department of Mathematics, College of Science, King Khalid University, Abha 61413, Saudi Arabia

**Keywords:** dusty fluid, variable viscosity, heat transfer, stratification, inclined irregular

## Abstract

The primary objective of the study is to explore the phenomena of dusty fluid flow through an inclined irregular channel under the impact of the transversely applied magnetic field of fixed strength. The density and viscosity of the working fluid are assumed to vary along with the height of the channel as it behaves as a replica of many real world mechanisms. Hence, a stratified dusty fluid through a channel that tilts to an angle θ is the main objective of the present study. The prescribed flow is mathematically modeled and it is approached numerically under two distinct boundary conditions. The finite difference technique is employed to discretize the system of equations and solved using the Thomas algorithm. The velocity and temperature fields are discussed for different pertinent parameters which influence the flow. The friction factor and heat transfer rate are discussed as it has been a subject of interest in recent decades. The results show that the stratification decay parameter leads to enhancement in the momentum of the fluid flow. The temperature field is found to be higher in the convective boundary than the Navier slip boundary.

## 1. Introduction

Thermal diffusion is a natural mechanism that occurs due to variation in temperature within the system. The phenomenon of heat transfer on dusty fluid flows is known to have abundant applications in many technological devices and industrial processes namely underground heat pump systems, grain storage, insulation materials, thermal design of buildings, commercial refrigeration, float glass production, etc. Initially, the mathematical model for the dusty fluid flows has been formulated by Saffman [1] to examine the characteristics of dusty gas. With this initiation, many researchers [2,3,4,5] continued their investigation on dusty fluid flows and on heat transfer analysis [6,7,8,9]. Manjunatha et al. [10] have focused on the effects of stretching parameter and heat transfer phenomenon on dusty fluid flow. Thermal radiation effects of non-Newtonian fluid that comprises the particle suspension through a planar channel on EMHD peristaltic propulsion have been studied theoretically by Bhatti et al. [11]. More recently, Rafiq et al. [12] have analyzed the nonlinear thermal radiation effect of a boundary layer dusty fluid flow and they have examined air and water in both pure and dusty forms.

During the past few decades, the heat transfer effect on magnetohydrodynamic (MHD) fluid flows are attractive from both mathematical and physical standpoints as it attributes that liquid metals occur either in nature or industries are electrically conducting. Such flow analyzes have considerable applications in engineering and technological problems including MHD generators, power generation systems, high temperature plasma, etc. Ibrahim [13] has employed the IV order scheme of Runge-Kutta Fehlberg accompanied by a shooting technique to determine the induced magnetic and convective heating impact on the flow and heat transfer of upper convected Maxwell fluid past a stretching sheet. This field has become more exigent to vast researchers and a few findings are included here. In the study of shock wave propagation of the dusty gas flow, the applied magnetic field reduces the compressibility in the flow field and thus weakens the shock strength. Vast research on the fluid flow under the influence of magnetic fields are available in recent years and few among them are [14]. The dust and nanoparticles impact the MHD fluid flow such that the heat transfer rate and friction factor can be improved with the addition of the particles and by strengthening the magnetic field [15]. Rehman et al. [16] have also confirmed that Casson dusty nanofluid exhibit low friction factor and heat transfer rate for higher magnetic field and Casson fluid parameter. With the enhancement of volume concentration of dust particles, the velocity reduces and the temperature raises [17]. A few recent research works that contributed to the MHD dusty flow are included here [18,19].

The heat transfer analysis of MHD fluid flows through a porous medium is significant since the porous matrix amplifies the process of heating/cooling and prevents heat loss as in a heat exchanger. The fluid momentum can be reduced remarkably when the flow is through a porous matrix [20]. The closed form of solution for heat and mass transfer effects of an electrically conducting MHD viscoelastic fluid through a porous medium has been given by Nayak et al. [21]. Metri et al. [22] have explained the effect of viscous dissipation and non-uniform heat source/sink on MHD flow and heat transfer characteristics of viscoelastic fluid over a permeable stretching sheet. Analyzing dusty fluid flows through different geometries have received considerable interest by many researchers [2,23,24,25]. In particular, inclined channels have gained much attention by investigators [26,27] since this geometry is very close to practical situations like a solar collector. Previous studies restricted their discussions by concerning flat or regular walls. Also, the study of heat transfer in irregular channels is more significant in many engineering problems. The phenomena of fluid through wavy boundaries are required in various areas such as film vaporization in combustion channels, transpiration cooling of rocket boosters, etc. Sivaraj and Rishikumar [28,29] have constructed a mathematical model to explain the thermal diffusion on unsteady MHD Couette flow of a viscoelastic dusty fluid through an irregular porous channel. When the dusty fluid flow is considered through the rotating disk, surface cooling is achieved by enhancing the rotation number [30].

The studies discussed above are focused mainly on flows with constant physical properties. However, the physical properties like density and viscosity of the dusty fluid may vary significantly due to changes in temperature. In recent years, understanding the instability phenomena that accompany the transportation of the fluid has been the area of interest in many real world problems. In the designs that purify the contaminated groundwater due to waste repositories have been major studies to reduce the health risks to humans. The viscosity and density of the underground fluid vary and depend on the temperature and porosity of the medium. Also, a major difference in the velocity of the fluid is noticed when the constant and variable viscosity is taken into account [31]. Lai and Kulacki [32] have used a theoretical model to examine the effects of temperature dependant viscosity along a vertical surface in a saturated porous medium. The influence of temperature, magnetic field, and porosity of the medium on the fluid flow through inclined channel have been presented by Chakraborty and Medhi [33]. It is evident from the study of Nazeer et al. [14] that variable viscosity reduces the momentum of the fluid flow and the trend is reversed for variable pressure.

Therefore, in view of the recent developments in various technological fields, this article is aimed to formulate the effects of variable pressure and variable viscosity of MHD dusty fluid flow over an inclined wavy channel through a porous medium. This model is studied under two different conditions, that is, convective and Navier slip boundaries which are found in various engineering problems. From the literature review, research on the fluid flow with variable pressure and viscosity is very narrow and this particular model is not studied before as per the authors’ knowledge. Therefore, the main objective of the article is concentrated on the examination of a tilted irregular channel under variable pressure and viscosity. The numerical solution is obtained and the influence of emerging parameters on velocity profiles and temperature distribution have been analyzed. In order to visualize the various phenomenon of the flow, the contours of velocity and temperature fields are plotted. Further, the variations of skin-friction coefficient and the local Nusselt number are tabulated in view of engineering applications.

## 2. Mathematical Model

The unsteady, laminar flow of an electrically conducting viscoelastic dusty fluid through an infinitely long irregular channel in a porous medium is considered for the analysis. The irregular channel is inclined at an angle θ. The *x*-axis overlaps with the direction of the flow and the magnetic field of fixed strength $B_{0}$ is applied normally to it (refer to Figure 1). For the present study, the dusty fluid is supposed to be as follows:The dust particles are distributed uniformly throughout the fluid flow.The number density of dust particles (*N*) is fixed.The dust particles are solid sphere shaped, identical and symmetrical in size.The nature of dust particles is elastic and electrically non-conducting.Except fluid density and viscosity all other physical properties are constant.

Due to temperature variation within the fluid medium, variation in fluid density and hence in viscosity occurs along the height of the channel [33] which is given by
$$\rho =\rho_{0}e^{-\alpha \left(\frac{2y}{h}+1\right)}\;\;\;\;\mathrm{and}\;\;\;\;\mu =\mu_{0}e^{-\alpha \left(\frac{2y}{h}+1\right)},$$
where $\rho_{0}$ and $\mu_{0}$ are respectively the coefficients of density and viscosity along the irregular wall. The flow is due to the pressure difference [33] which is given as
p(x,y,t)=ρg(xsinθ−ycosθ)+ρxd,
where *d* is a function of *t* alone and θ is the inclination of the channel. Taking these assumptions into the consideration, the governing equations based on Prandtl boundary layer theory [34] are as follows:(1)$$\frac{\partial u}{\partial t}=g\sin\theta -\frac{1}{\rho }\frac{\partial p}{\partial x}+\frac{1}{\rho }\frac{\partial }{\partial y}\left(\mu \frac{\partial u}{\partial y}\right)-\frac{\sigma B_{0}^{2}u}{\rho }+\nu \lambda \frac{\partial }{\partial t}\left(\frac{\partial^{2}u}{\partial y^{2}}\right)+\frac{kN}{\rho }\left(v-u\right)-\frac{\nu u}{\eta },$$
(2)$$\frac{\partial p}{\partial y}+g\cos\theta =0,$$
(3)$$\frac{\partial v}{\partial t}=\frac{k}{m}\left(u-v\right),$$
(4)$$\frac{K_{T}}{\rho C_{p}}\frac{\partial^{2}T}{\partial y^{2}}=\frac{\partial T}{\partial t}-\frac{\nu }{C_{p}}\left(1+\lambda \frac{\partial }{\partial t}\right){\left(\frac{\partial u}{\partial y}\right)}^{2}.$$

The prescribed dusty fluid flow is evaluated for the following two cases which are mentioned below:


**Case 1**


The convective boundary conditions respectively $-K_{T}\frac{\partial T}{\partial y}=h_{f}\left[T_{0}-T+\left(T_{w}-T_{0}\right)\varepsilon e^{-nt}\right]$ lower than $-K_{T}\frac{\partial T}{\partial y}=h_{f}\left[T_{w}-T+\left(T_{w}-T_{0}\right)\varepsilon e^{-nt}\right]$ represents the irregular $\left(y=\epsilon \cos\left(k_{1}x\right)\right)$ and flat $\left(y=\frac{h}{2}\right)$ walls which maintain the temperature $T_{w_{1}}$ and $T_{w_{2}}$ respectively. The conditions of velocity fields and temperature at the boundaries are given by

i.when t=0, u=0=v, $T=T_{0}$, for $y\;\in \;\left(\epsilon \;\cos\left(k_{1}x\right),\frac{h}{2}\right)$ii.when t>0, u=0=v, $-K_{T}\frac{\partial T}{\partial y}=h_{f}\left[T_{0}-T+\left(T_{w}-T_{0}\right)\varepsilon e^{-nt}\right]$, for $y=\epsilon \;\cos(k_{1}x)$


(5)
$$u=0=v,\;\;\;-K_{T}\frac{\partial T}{\partial y}=h_{f}\left[T_{w}-T+\left(T_{w}-T_{0}\right)\varepsilon e^{-nt}\right]\;\;\;\mathrm{for}\;\;\;y=\frac{h}{2}.\;\;\;\;\;\;\;\;\;\;$$



**Case 2**


In this case, the velocity near the boundary is specified using the Navier slip condition. The boundary conditions for velocity and temperature fields in the dimensional form are provided below:i.when t≤0, u=0=v, $T=T_{0}$, for $y\;\in \;\left(\epsilon \;\cos\left(k_{1}x\right),\frac{h}{2}\right)$ii.when t>0, $\frac{h\beta }{2}\left(\frac{\partial u}{\partial y}\right)=u$, v=0, $T=T_{0}$ for $y=\epsilon \;\cos(k_{1}x)$

(6)$$\frac{h\beta }{2}\left(\frac{\partial u}{\partial y}\right)=u,\;\;\;v=0,\;\;\;T=T_{w}\;\;\mathrm{for}\;\;\;y=\frac{h}{2}.\;\;\;\;\;\;\;\;\;\;\;\;\;\;\;\;\;\;\;\;\;\;\;\;\;\;\;\;\;\;\;\;\;\;\;\;\;\;\;\;\;\;\;\;\;\;\;\;\;\;\;\;\;\;\;\;\;\;\;\;\;\;\;\;\;\;\;$$

The following parameters are introduced to reduce the governing equations to dimensionless form:(7)$$\begin{array}{c} y^{*}=\frac{2y}{h},\;\;\;\;x^{*}=\frac{2x}{h},\;\;\;\;u^{*}=\frac{u}{u_{0}},\;\;\;\;v^{*}=\frac{v}{u_{0}},\;\;\;t^{*}=\frac{2u_{0}t}{h},\;\;\;\epsilon^{*}=\frac{2\epsilon }{h},\;\;\;d^{*}=\frac{dh}{2u_{0}^{2}},\;\;\;\\ T^{*}=\frac{T-T_{0}}{T_{w}-T_{0}},\;\;\;\;n^{*}=\frac{nh}{2u_{0}},\;\;\;\kappa^{*}=\frac{h}{2\sqrt{\eta }}. \end{array}$$

The non-dimensional of Equations (Equation 3)–(Equation 6), after dropping the * are as follows,
(8)$$\frac{\partial u}{\partial t}=-d-\frac{\alpha }{R_{e}}\left(\frac{\partial u}{\partial y}\right)+\left(\frac{1}{R_{e}}-V_{e}\frac{\partial }{\partial t}\right)\left(\frac{\partial^{2}u}{\partial y^{2}}\right)+\frac{l}{\tau }\left(v-u\right)-\frac{H_{a}^{2}}{R_{e}}u-\frac{\kappa^{2}}{Re}u,$$
(9)$$\frac{\partial v}{\partial t}=\frac{1}{\tau }\left(u-v\right),$$
(10)$$\frac{\partial^{2}T}{\partial y^{2}}=R_{e}P_{r}\frac{\partial T}{\partial t}-E_{c}P_{r}\left(1-V_{e}R_{e}\frac{\partial }{\partial t}\right){\left(\frac{\partial u}{\partial y}\right)}^{2},$$
where *d* is supposed to be $d_{0}\cos\left(d_{1}+d_{2}t\right)$.

The dimensionless form of the boundary conditions (Equation 7) and (Equation 8) are respectively given below:i.when t=0,u=0=v=T, for y∈(b,1)ii.when $t>0,\;\;\;u=0=v,\;\;\frac{\partial T}{\partial y}=\frac{B_{i}}{2}\left(T-\varepsilon e^{-nt}\right)\mathrm{for}\;\;y=b$
(11)$$u=0=v,\;\;\frac{\partial T}{\partial y}=\frac{B_{i}}{2}\left(T-\varepsilon e^{-nt}-1\right)\;\;\mathrm{for}\;\;y=1,$$


where $b=\varepsilon \cos(\lambda_{1}x)$ and

i.when t≤0,u=0=v=T for y∈(b,1)ii.when $t>0,\;\;\beta \left(\frac{\partial u}{\partial y}\right)=u$, v=0, T=0
fory=b
(12)$$\beta \left(\frac{\partial u}{\partial y}\right)=u,\;\;\;v=0,\;\;\;T=1\;\;\;\mathrm{for}\;\;y=1,\;\;\;\;\;\;\;\;\;\;\;\;\;\;\;\;\;\;\;\;\;\;\;\;\;\;\;\;\;\;\;\;\;\;\;\;\;\;\;\;\;\;\;\;\;\;\;\;\;\;\;\;\;\;\;\;\;\;\;\;\;\;\;\;\;\;\;\;\;\;\;\;\;\;\;\;\;\;\;\;\;$$


where β is the Navier slip parameter.

## 3. Numerical Solution

A numerical approach to the mathematical model is preferable to an analytical approach, as it conquers the difficulties of time consumption, solving techniques, etc. Hence, the prescribed flow is analyzed by a finite difference scheme. The approximations of coupled equations of fluid and dust velocities (Equation 10) and (Equation 11) and temperature (Equation 12) are given by
(13)$$\begin{array}{c} C_{1}\left(u_{i+1,j+1}+u_{i-1,j+1}\right)+C_{2}u_{i,j+1}=C_{4}u_{i-1,j}+C_{3}\cos\left(d_{1}+jh_{2}d_{2}\right)+C_{5}u_{i,j}+C_{6}u_{i+1,j}+C_{7}v_{i,j}, \end{array}$$
(14)$$\begin{array}{c} v_{i,j+1}=C_{8}u_{i,j}+C_{9}v_{i,j},\\ C_{10}T_{i,j+1}=C_{11}\left(T_{i+1,j}+T_{i-1,j}\right)+C_{12}T_{i,j}+C_{13}\left(u_{i+1,j}u_{i,j+1}-u_{i+1,j}u_{i+1,j+1}\right. \end{array}$$
(15)$$\begin{array}{c} \;\;\;\;\;\;\;\;\;\;\left.+u_{i+1,j+1}u_{i,j}-u_{i,j+1}u_{i,j}\right)+C_{14}{\left(u_{i+1,j}-u_{i,j}\right)}^{2}. \end{array}$$

The corresponding numerical approximation of boundary conditions (Case: 1) (Equation 13) are
i.when t=0, $u_{i,0}=0=v_{i,0}=T_{i,0},\;\;\;\forall i$,ii.when $t>0,\;\;u_{0,j}=0=v_{0,j},\;\;\;T_{1,j}=C_{15}T_{0,j}+C_{16},$



$\;\;\;\;\;\;\;\;\;\;u_{n_{1},j}=u_{p}=v_{n_{1},j}\;\;\;T_{n_{1}+1,j}=C_{15}T_{n_{1},j}+C_{16}-\frac{h_{1}B_{i}}{2},\;\;\;\forall j,$



and the numerical discretization of the boundary conditions (Case: 2) (Equation 14) are

i.when t=0, $u_{i,0}=0=v_{i,0}=T_{i,0},\;\;\;\forall i$ii.when $t>0,\;\;u_{0,j}=\left(1+\frac{h_{1}}{\beta }\right)u_{1,j},\;\;v_{0,j}=0=T_{0,j},$



$\;\;\;\;\;\;\;\;\;\;\;\;\;\;\;\;\;\;\;\;\;\;u_{n_{1},j}=\left(1+\frac{h_{1}}{\beta }\right)u_{n_{1}+1,j},v_{n_{1},j}=0,\;\;T_{n_{1},j}=1,\;\;\;\forall j,$



where


$$\begin{array}{c} C_{1}=\frac{Ve}{h_{1}^{2}},\;\;\;C_{2}=1-2C_{1},\;\;\;C_{3}=-d_{0}h_{2},C_{4}=C_{1}+\frac{h_{2}}{R_{e}h_{1}^{2}},\;\;\;C_{5}=C_{2}+\frac{\alpha h_{2}}{R_{e}h_{1}}-\frac{2h_{2}}{R_{e}h_{1}^{2}}-\frac{\kappa^{2}h_{2}}{R_{e}}-\frac{H_{a}^{2}h_{2}}{R_{e}}-\frac{lh_{2}}{\tau },\\ C_{6}=C_{4}-\frac{\alpha h_{2}}{R_{e}h_{1}},\;\;\;C_{7}=\frac{lh_{2}}{\tau },\;\;\;C_{8}=\frac{h_{2}}{\tau },\;\;\;C_{9}=1-C_{8},\;\;\;C_{10}=\frac{R_{e}P_{r}}{h_{2}},\;\;\;C_{11}=\frac{1}{h_{1}^{2}},\;\;\;C_{12}=C_{10}-2C_{11},\\ C_{13}=\frac{2V_{e}E_{c}P_{r}R_{e}}{h_{1}^{2}h_{2}},\;\;\;C_{14}=C_{13}+\frac{E_{c}P_{r}}{h_{1}^{2}},\;\;\;C_{15}=1+\frac{B_{i}h_{1}}{2},\;\;\;C_{16}=-\frac{h_{1}B_{i}\varepsilon e^{-njh_{2}}}{2}. \end{array}$$


Here $h_{1}$ and $h_{2}$ are the mesh size respectively along and normal to the flow directions. Also, $n_{1}$ and $n_{2}$ are the number of grids accordingly along *x* and *y* axes and it is based on the boundaries in which the flow is analyzed. In the current problem, the numerical algorithm was executed for different values of $h_{1}$ and $h_{2}$. Whenever the magnitude of the grid points is very small, the results are determined to be in close agreement. The convergence criteria of the current solutions are defined as
$$|u^{m+1}-u^{m}\left|<0.0000001\;\;\;\mathrm{and}\;\;\;\right|u^{m+1}-u^{m}|<0.0000001,$$
where *m* and m+1 indicate the number of iterations. It is essential to mention that, the solutions obtained are ascertained to be identical for any values of $h_{2}\leq 0.02$ which confirms the convergence criteria of the results. Thus, all the outcomes of the flow phenomena are extracted by choosing $h_{2}=0.02$ and the results established in graphical form are drawn at x=0.8.

The analysis of shear stress and heat transfer rate near the wall have been researched for centuries. Over the last few decades, knowledge of these factors become more focal area, especially in aerospace sectors which bridges to improvise the quality while designing the equipment.

### 3.1. Skin-Friction ($C_{f}$)

The friction between a moving dusty fluid and the inclined channel i.e., wall shear stress is analyzed through skin-friction and is given by
$$\begin{array}{cc} C_{f}=\frac{2\tau_{w}}{\rho_{\infty }u_{0}^{2}},\;\;\mathrm{where}\;\;\tau_{w}=\mu {\left(\frac{\partial u}{\partial y}\right)}_{y=0} & \end{array}$$
is the shearing stress. The skin-friction coefficient in non-dimensional form is:$$\begin{array}{cc} C_{f}=\frac{e^{-\alpha }}{Re}{\left(\frac{\partial u}{\partial y}\right)}_{y=0}. & \end{array}$$

### 3.2. Nusselt Number (Nu)

The comparison of heat transfer rate between the conduction and convection of fluid can be analyzed by the Nusselt number. The dimensional heat transfer coefficient
$$\begin{array}{cc} Nu=-\frac{xq_{w}}{\lambda_{\infty }\left(T_{w}-T_{\infty }\right)},\;\;\mathrm{where}\;\;q_{w}=-\lambda {\left(\frac{\partial T}{\partial y}\right)}_{y=0} & \end{array}$$
is wall heat flux. When non-dimensional variables are used, the Nusselt number reduced to
$$\begin{array}{cc} Nu=x{\left(\frac{\partial T}{\partial y}\right)}_{y=0}. & \end{array}$$

## 4. Results and Description

This section provides graphical solutions and descriptions for the same to study the behavior of dusty fluid flow and thermal diffusion under the impact of various parameters involved. Figure 2, Figure 3, Figure 4, Figure 5, Figure 6, Figure 7, Figure 8 and Figure 9 demonstrate the velocity u,v and temperature *T* distributions against a few physical parameters which have a strong influence on the dusty fluid flow subjected to two different boundary conditions mentioned above. Alike nature in the variation of u,v, and *T* is noticed in both cases. However, the magnitude differs with respect to adopted boundary conditions. Hence the physical interpretations and consequences thus discussed below for the flow phenomena over distinct parameters are going well with cases 1 and 2. In the figures the case 1 and 2 figures are referred as (a) and (b) respectively. Figure 2a,b describe the development of velocity profiles for stratification decay parameter (α). It is seen that increasing the strength of the stratification decay parameter, increases the velocities of dusty fluid flow. Here, α is related inversely to the viscosity of the dusty fluid. Therefore, retarding force occurs in the fluid flow due to viscous force reduction, and thus velocity increases.

It is apparent from Figure 3a,b that fluid and dust velocities notably increase for higher values of porous permeability parameter. Since the porous medium reduces the drag force, hence velocity enhances. It can be explained theoretically as, permeability of porous medium is inversely proportional to porous parameter, i.e., $\kappa =\frac{1}{\sqrt{\eta }}$. Raising the value of the porous parameter decelerates the flow or equivalently saying that high momentum in the flow can be achieved by enhancing the permeability of the porous medium. Therefore, it is concluded that the resistive force thus developed in dusty fluid flow can be reduced in a high permeability medium.

Fluid and dust phase velocities strictly increase for rising Reynolds number ($R_{e}$) and it is illustrated in Figure 4a,b. This phenomenon is observed since the Reynolds number depends inversely to viscous force. Since relative internal movement in the fluid flow improves when $R_{e}$ grows. Simultaneously, a reduction in the resistive force developed in the momentum due to a decline in the viscosity of the fluid. It is clear that a higher magnitude of $R_{e}$ weakens the viscous force and strengthens the inertial force in the flow and thus resulting in higher velocities.

Figure 5a,b elucidate the effect of the magnetic field on the velocity distribution across the channel. Here, the transversely imposed magnetic field produced a kind of drag or resistive force namely Lorentz force which significantly retards the transport phenomena, consequently velocity fields decrease. However, it is evident from Figure 6a,b that the impact is reversed in temperature fields i.e., an increase in the magnitude of the Hartmann number enhances temperature profiles. This may happen due to a reduction in the thickness of the momentum boundary layer and a thickening in the thermal boundary layer.

Inspection of Figure 7a,b exhibit the response in temperature for varying values of Eckert number ($E_{c}$). The temperature is identified as an increasing function with respect to the high Eckert number. This parameter is defined as the ratio of the kinetic energy of the flow to the enthalpy difference near the boundary layer. It indicates the amount of kinetic energy converted into internal energy during work done against stresses of the viscous fluid. Therefore, kinetic energy in the channel has been converted into heat through viscous dissipation for a greater Eckert number. Physically, a rise in the value of $E_{c}$ corresponds to the cooling of walls of the inclined channel. Also, the thermal boundary layer becomes thicker for greater $E_{c}$ correspondingly increases the viscous dissipative heat energy which results into a rise in the temperature of the dusty fluid.

Figure 8a,b clear that with an increment in the Prandtl number ($P_{r}$), there is a reduction in temperature profiles. This phenomenon occurs so, that with the increment in $P_{r}$, the thermal layer becomes thinner and the capability of heat transfer amplifies. The fluids with large $P_{r}$ have relatively low thermal conductivity and as a consequence, reduction in conduction develops and thereby temperature decreases. $P_{r}$ values taken in this study are 0.2,0.63, and 0.71 which are respectively related to the mixture of noble gases, oxygen, and air.

Figure 9 depicts the temperature profile against different values of the Biot number. The ratio of the internal heat transfer resistances to the thermal resistance in the boundary layer describes the Biot number, therefore convective heat transfer to the dusty fluid raises for large values of the Biot number. Hence, Figure 9 reflects that with a greater Biot number, the magnitude of the temperature is suppressed. It is crucial to mention that the temperature field in Figure 7a, Figure 8a and Figure 9 are decreasing from the oscillatory boundary and increases from the middle of the channel. It happens because the convective boundaries are assumed to vary exponentially.

The skin-friction coefficient ($C_{f}$) and heat transfer rate (Nu) which are the main concern of engineers and industrialists are illustrated in Table 1 for different physical parameters. The lesser value of $C_{f}$ indicates higher motion in the fluid flow which is expected in any better model. It can be attained by choosing a higher magnitude in the stratification decay parameter (α), Reynolds number ($R_{e}$), and permeability of porosity (η). Table 1 reveals that friction factor possibly decline by boosting the values of α, $R_{e}$ and η. Additionally, it is observed that $C_{f}$ is comparatively more in case 1 than in case 2 for these parameters. Whereas, the strength of the magnetic field has to shrink in order to establish low shear stress in the flow and $C_{f}$ is relatively high in case 2. A high heat transfer rate is achieved by increasing the value of the Hartmann number ($H_{a}$), Eckert number ($E_{c}$), and Biot number. Contrastingly, the heat transfer rate reduces whenever the Prandtl number ($P_{r}$) elevates. Further, the magnitude of Nu is reasonably more in case 1 while varying $H_{a}$ and $P_{r}$ and comparatively superior in case 2 when $E_{c}$ is raised.

## 5. Conclusions

A numerical technique has been adopted to describe viscoelastic dusty fluid flow through an inclined channel for convective and Navier slip boundaries. Discussion on flow phenomena under the effect of various parameters has been provided. The impact of parameters on shear and heat transfer rates are also computed. Based on this study, the following conclusions are drawn:The dusty fluid flow approaches high velocity when the stratified decay parameter, Reynolds number, and permeability of porous medium increase, whereas the effect is contrary for Hartmann number.Noticeable enhancement in temperature profile is observed for raising values of Eckert number, Hartmann number and reducing the value of Prandtl number and Biot number.Velocity fields of dusty fluid phase in case 1 is comparatively higher than in case 2 irrespective of variation in flow pertinent parameters.Temperature distribution is found to be more in the case of convective boundary than in Navier slip boundary, i.e., case 2.Increment in stratified decay parameter, Reynold number, and permeability of porosity yield reduction in friction factor, and the opposite effect is experienced with increment in Hartmann number.The heat transfer rate is magnified by boosting the values of $H_{a}$, $E_{c}$ number, and $B_{i}$ and by reducing the magnitude of $P_{r}$.

The current study totally focused on the stratified dusty fluid. This study has further scope if it will be modeled to a dusty nanofluid or with non-Newtonian fluid that has high real world applications.

## Figures and Tables

**Figure 1 nanomaterials-12-03309-f001:**
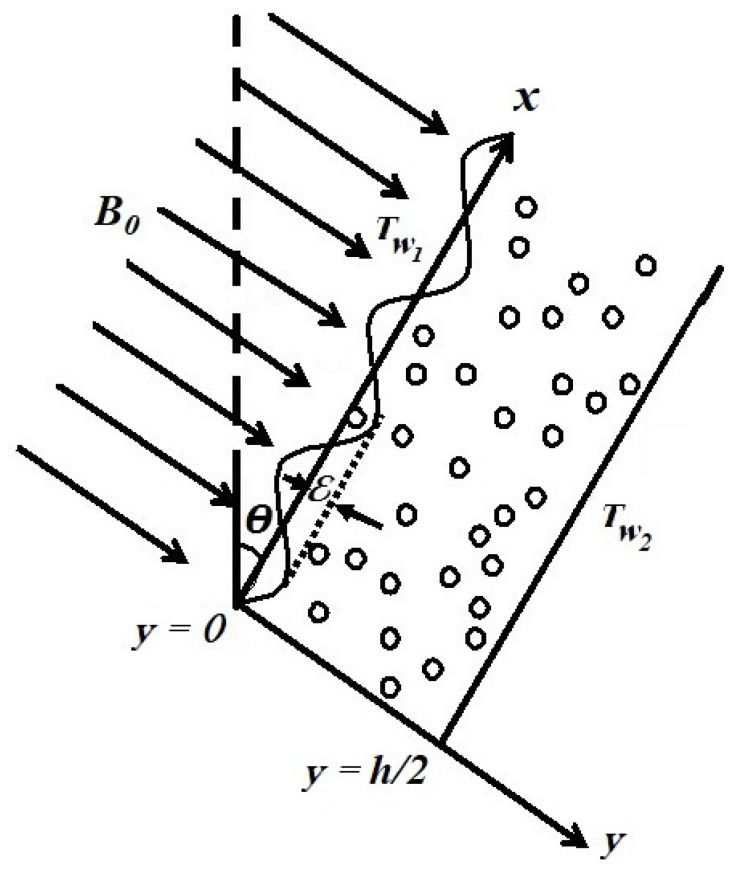
Flow geometry.

**Figure 2 nanomaterials-12-03309-f002:**
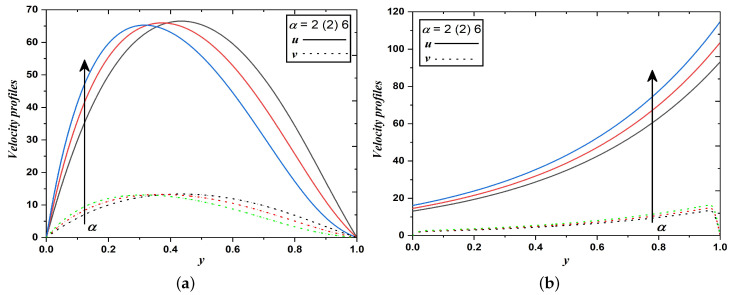
Velocity profiles when stratification decay parameter varies (Case 1 & 2).

**Figure 3 nanomaterials-12-03309-f003:**
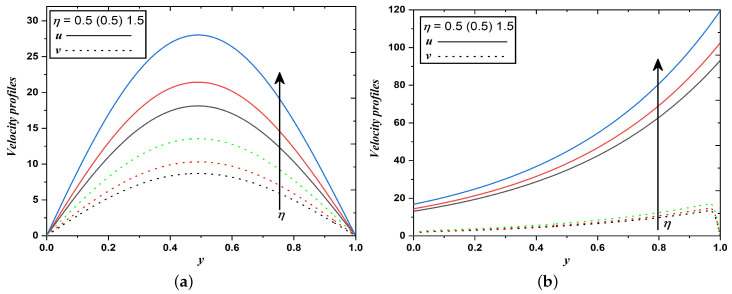
Velocity profiles when the permeability of porous medium varies (Case 1 & 2).

**Figure 4 nanomaterials-12-03309-f004:**
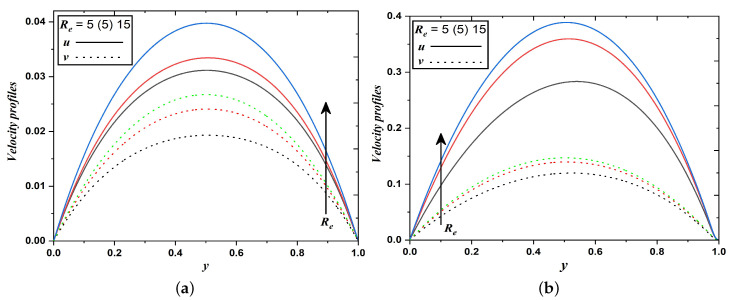
Velocity profiles when Reynolds number varies (Case 1 & 2).

**Figure 5 nanomaterials-12-03309-f005:**
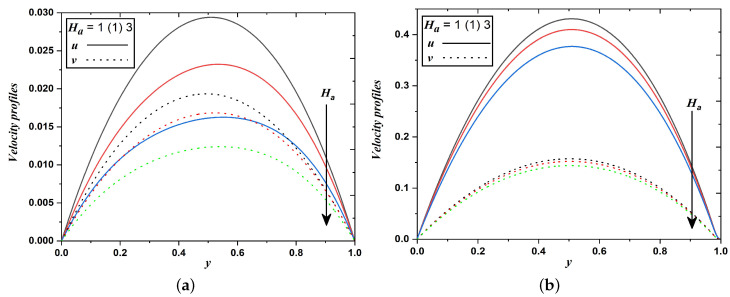
Velocity profiles when Hartmann number varies (Case 1 & 2).

**Figure 6 nanomaterials-12-03309-f006:**
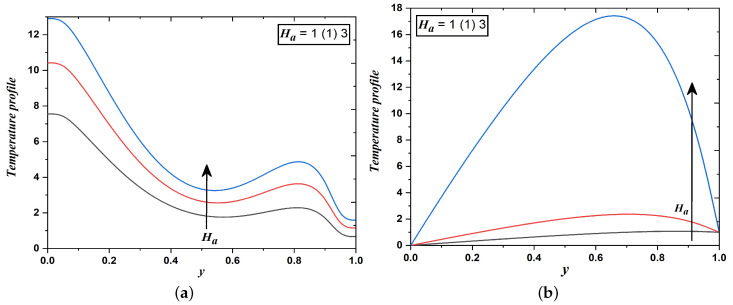
Temperature field when Hartmann number varies (Case 1 & 2).

**Figure 7 nanomaterials-12-03309-f007:**
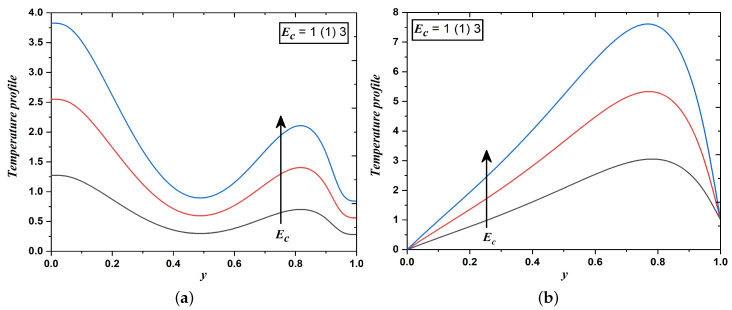
Temperature field when Eckert number varies (Case 1 & 2).

**Figure 8 nanomaterials-12-03309-f008:**
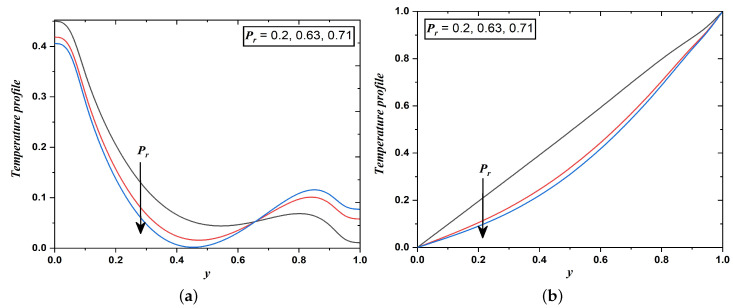
Temperature field when Prandtl number varies (Case 1 & 2).

**Figure 9 nanomaterials-12-03309-f009:**
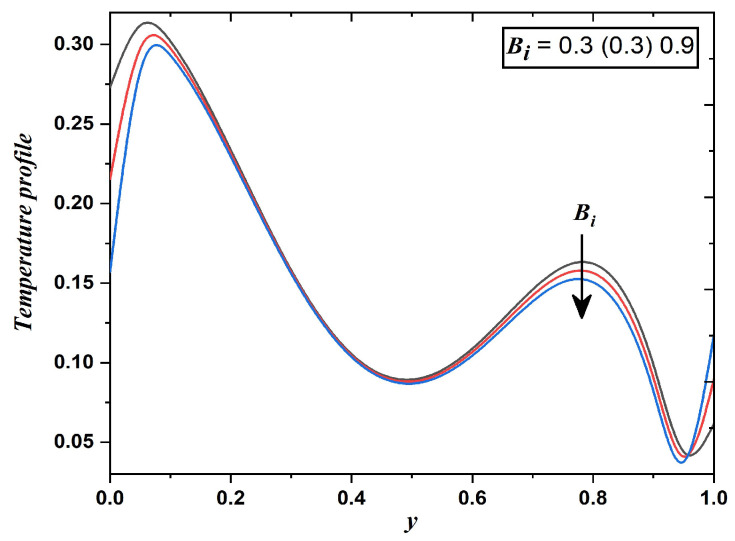
Temperature field when Biot number varies (Case 1).

**Table 1 nanomaterials-12-03309-t001:** The numerical value of skin-friction and Nusselt number for distinct physical parameters.

Parameters							Friction Factor		Nusselt Number	
α	$R_{e}$	η	$H_{a}$	$P_{r}$	$E_{c}$	$B_{i}$	**Case 1**	**Case 2**	**Case 1**	**Case 2**
2							1.655429831	1.522853027	-	-
4	5	0.5	2	0.63	2	0.3	0.267180727	0.051919641	-	-
6							0.047586102	0.042784836	-	-
	5						0.05630821	0.051046954	-	-
2	10	0.5	2	0.63	2	0.3	0.041519842	0.011958089	-	-
	15						0.012577104	0.006690786	-	-
		0.5					0.072013672	0.184309882	-	-
2	5	1	2	0.63	2	0.3	0.35154075	0.093963489	-	-
		1.5					0.50068426	0.070470312	-	-
			1				0.00001230	0.014340811	2.668474288	0.16881461
2	5	1	2	0.63	2	0.3	0.000126263	0.017545688	3.097579452	0.471514472
			1.5				0.018380056	0.043992287	3.389508739	3.781870901
				0.2			-	-	0.747356087	0.097687653
2	5	1	2	0.63	2	0.3	-	-	0.728959459	0.049202599
				0.71			-	-	0.855920223	0.041396851
					1		-	-	0.079504801	0.410427683
2	5	1	2	0.63	2	0.3	-	-	0.160255859	0.721319137
					3		-	-	0.241006918	1.032210592
						0.3	-	-	0.378036935	-
2	5	1	2	0.63	2	0.6	-	-	0.732270204	-
						0.9	-	-	1.093369217	-

## Data Availability

Not applicable.

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
