# Peer review of "Heat Transfer of Magnetohydrodynamic Stratified Dusty Fluid Flow through an Inclined Irregular Porous Channel"

_nanomaterials, 2022, doi:10.3390/nano12193309_

Round 1

Reviewer 1 Report

Comment 1:  The abstract is well written, but it is better to include methodology and novel point of study.

Comment 2: Introduction contain very old study. I suggest to add recently published articles on relevant to your topic. More related references should be added to the introduction section. Must include real life applications of elaborated problem.

Comment 3:  Must update last paragraph of introduction section which contains the motivation of study, the problem context, and add specific objectives.

Comment 4:  What is the added value of this study? What are the differences, benefits, drawbacks? There is a lack of clear explanations and comparison with state of the art.

Comment 5: There are some recent study on nanofluids and published works. The authors should cite all of them. The Casson Dusty Nanofluid: Significance of Darcy–Forchheimer Law, Magnetic Field, and Non-Fourier Heat Flux Model Subject to Stretch Surface. Micropolar Dusty Fluid: Coriolis Force Effects on Dynamics of MHD Rotating Fluid When Lorentz Force Is Significant. Significance of Dust Particles, Nanoparticles Radius, Coriolis and Lorentz Forces: The Case of Maxwell Dusty Fluid. Computational Analysis for Bioconvection of Microorganisms in Prandtl Nanofluid Darcy–Forchheimer Flow across an Inclined Sheet.

Comment 6: There are some equations coming from other paper's theories and published works. The authors should cite all of them. Also cite reference in the variable viscosity mathematically expression.

Comment 7: All figures should be discussed from the physical implications, significance justifications point of view, hence, the discussion of results section needs to be improved.

Comment 8:  For validation of your results, must compared your results with already published data for limited cases.

Comment 9: Please expand the conclusions in relation to the specific goals and the future work.

Author Response

(1) The abstract is well written, but it is better to include methodology and a novel point
of study.
Response : The abstract is revised.
(2) Introduction contain very old study. I suggest to add recently published articles on
relevant to your topic. More related references should be added to the introduction
section. Must include real life applications of elaborated problem.
Response : Recent articles and real world applications are included in the
revised manuscript.
(3) Must update last paragraph of introduction section which contains the motivation of
study, the problem context, and add specific objectives.
Response : It is incorporated in the manuscript.
(4) What is the added value of this study? What are the differences, benefits, draw-
backs? There is a lack of clear explanations and comparison with state of the art.
Response : It is included in the article.
(5) There are some recent study on nano fluids and published works. The authors should
cite all of them. The Casson Dusty Nano fluid: Significance of Darcy-Forchheimer
Law, Magnetic Field, and Non-Fourier Heat Flux Model Subject to Stretch Surface.
Micropolar Dusty Fluid: Coriolis Force Effects on Dynamics of MHD Rotating Fluid
When Lorentz Force Is Significant. Significance of Dust Particles, Nanoparticles
Radius, Coriolis and Lorentz Forces: The Case of Maxwell Dusty Fluid. Compu-
tational Analysis for Bioconvection of Microorganisms in Prandtl Nano fluid Darcy-Forchheimer Flow across an Inclined Sheet.
Response : It is updated in the revised article.
(6) There are some equations coming from other paper's theories and published works.
The authors should cite all of them. Also cite reference in the variable viscosity
1
2
mathematically expression.
Response : References are added in the revised article.
(7) All  figures should be discussed from the physical implications, significance justifications point of view, hence, the discussion of results section needs to be improved.
Response : It is improvised in the revised article.
(8) For validation of your results, must compared your results with already published data for limited cases.
Response : Validation of the result is done by carrying out a grid independence test and by defining strong convergence criteria. We couldn't find a proper article to do a comparative study as a limiting case. So, a
comparative study is not included in the revised article.
(9) Please expand the conclusions in relation to the specific goals and the future work.
Response : The conclusions are improvised and future scope also included
in the revised manuscript.

Reviewer 2 Report

Unfortunately, the present work is not contributing to any field of applied mathematics. In this paper, a standard technique is applied to a problem that is very slightly different from a previously published problem. The research problem in the paper does not seem to be motivated by a clearly outlined research question and no physical insight is provided for this theoretical analysis.

Author Response

Unfortunately, the present work is not contributing to any  field of applied mathematics. In this paper, a standard technique is applied to a problem that is very slightly different from a previously published problem. The research problem in the paper does not seem to be motivated by a clearly outlined research question and no physical insight is provided for this theoretical analysis.
Response : The quality of the article has improved after incorporating the suggestions given by the reviewers.

Reviewer 3 Report

This work studied the heat transfer process of magnetohydrodynamic fluid flow through an inclined irregular porous channel. This topic is interesting for heat exchangers. However, some issues should be addressed before publication.

1 The novelty of this work should be enhanced.

2 How to verify the model used in this work?

3 More depth understanding about the heat transfer process of magnetohydrodynamic fluid should be discussed.

4 More attention should be paid on the paper writing for better reading.

Author Response

This work studied the heat transfer process of magnetohydrodynamic  fluid  ow through an
inclined irregular porous channel. This topic is interesting for heat exchangers. However,
some issues should be addressed before publication.
(1) The novelty of this work should be enhanced.
Response : The novelty is revised in the article.
(2) How to verify the model used in this work?
Response : The mathematical model developed is verified by defining
strong convergence criteria and by checking the grid independence test.
(3) More depth understanding about the heat transfer process of magnetohydrodynamic fluid should be discussed.
Response : Few more points are included in the revised article.
3
(4) More attention should be paid on the paper writing for better reading.
Response : Authors have taken extra care while revising the manuscript.

Round 2

Reviewer 1 Report

The authors address my comments with satisfactory changes so the present form is suitable for publication. Congratulation. 

Reviewer 3 Report

All comments have been addressed.